# Association between polygenic propensity for psychiatric disorders and nutrient intake

Avina K. Hunjan [1,2], Christopher Hübel [1,2,3], Yuhao Lin[1], Thalia C. Eley [1,2] & Gerome Breen [1,2✉]

Despite the observed associations between psychiatric disorders and nutrient intake, genetic studies are limited. We examined whether polygenic scores for psychiatric disorders are associated with nutrient intake in UK Biobank ($N = 163,619$) using linear mixed models. We found polygenic scores for attention-deficit/hyperactivity disorder, bipolar disorder, and schizophrenia showed the highest number of associations, while a polygenic score for autism spectrum disorder showed no association. The relatively weaker obsessive-compulsive disorder polygenic score showed the greatest effect sizes suggesting its association with diet traits may become more apparent with larger genome-wide analyses. A higher alcohol dependence polygenic score was associated with higher alcohol intake and individuals with higher persistent thinness polygenic scores reported their food to weigh less, both independent of socioeconomic status. Our findings suggest that polygenic propensity for a psychiatric disorder is associated with dietary behaviour. Note, nutrient intake was self-reported and findings must therefore be interpreted mindfully.

[1] Social Genetic & Developmental Psychiatry Centre, Institute of Psychiatry, Psychology and Neuroscience, King's College London, London, UK. [2] NIHR Biomedical Research Centre for Mental Health; South London and Maudsley NHS Trust, London, UK. [3] National Centre for Register-based Research, Department of Economics and Business Economics, Aarhus University, Aarhus, Denmark. ✉email: gerome.breen@kcl.ac.uk

Psychiatric disorders such as major depressive disorder (MDD), schizophrenia, and attention deficit hyperactivity disorder (ADHD) affect 20−25% of the population at any one time[1,2]. There are multiple reports of deficiencies in micronutrients, including vitamins B6 and B9 (i.e., folate) in ADHD and its symptoms[3]; B9 and B12 with worse negative symptoms in schizophrenia[4]; and vitamin B12 in obsessive-compulsive disorder (OCD) risk[5]. Other nutrient deficiencies are also common in individuals with a psychiatric diagnosis[6,7]. Taken together, these observations demonstrate the need for a more in-depth investigation of the co-occurrence between psychiatric disorders and alterations in dietary behaviour.

Both psychiatric disorders and nutrient intake are heritable. Psychiatric disorders are complex traits influenced by thousands of genetic variants, and genome-wide association studies (GWAS) have identified more than 300 independent genomic loci[8−11]. Twin analyses have also revealed strong heritability for total energy intake (twin-$h^2$ = 48%), macronutrients (35−45%; i.e., protein, carbohydrates, and fat), minerals (45%; includes calcium, iron, and potassium), and vitamins (21%; including vitamins A, C, D, E, and carotenoids, such as alpha and beta carotenes)[12]. Six GWAS of dietary intake have been performed[13−18], with the largest identifying 309 associated loci, including olfactory receptor associations with fruit and tea intake[18].

Despite (i) the observed phenotypic associations between psychiatric disorders and nutrient intake, and (ii) the heritability of both, little empirical attention has been given to understanding their genetic overlap. A recent UK Biobank study found significant genetic correlations between schizophrenia and two diet groups—one representing a meat-related diet and the other a fish and plant-related diet[19]. This study highlights the need for more genetic studies to better understand the relationship between psychiatric disorders and nutrient intake. This is clinically important because unhealthy dietary habits can impact physical and psycho-social health[20] imposing a further burden on individuals with a psychiatric disorder. Genetic studies could help determine whether the development of integrative treatment strategies that target behaviour alterations in the diet is needed to improve the long-term management of these disorders.

This study explores the association between polygenic scores for psychiatric disorders and self-reported nutrient intake using data from the UK Biobank, to determine whether polygenic propensity for a psycmhiatric disorder is associated with dietary behaviour. Specifically, we looked at polygenic scores for eight psychiatric disorders, including anorexia nervosa, schizophrenia, and MDD, as well as food addiction, persistent thinness, educational attainment, body mass index (BMI), and height, using systemic lupus erythematosus (lupus) as a negative control. We examined their association with the intake of fourteen nutrients derived from the UK Biobank 'Diet by 24-h recall' questionnaire.

Here, we showed that polygenic scores for seven psychiatric disorders and several behavioural and anthropometric traits significantly associate with self-reported nutrient intake on an average day. Polygenic scores for schizophrenia, bipolar disorder, and ADHD showed the highest number of significant associations with the intake of specific nutrients, whilst a polygenic score for autism spectrum disorder showed no associations. The relatively weaker OCD polygenic score showed the greatest effect sizes. In addition, contrary to our expectation, a higher BMI polygenic score was linked to a lower intake of fats and carbohydrates. Furthermore, we found further evidence to suggest pleiotropy of polygenic factors associated with educational attainment. Our findings encourage further research into the shared biological pathways and common environmental factors influencing nutrient intake and psychiatric disorders, in addition to behavioural and anthropometric traits.

## Results

**Descriptives**. Characteristics of the study participants are summarised in Table 1. From our sample of 163,619 participants, 73,853 (45.1%) were males and 89,799 (54.9%) were females. Participants were aged between 40 and 72 (mean age of 56). Forty percent ($n = 65,296$) of our sample completed the questionnaire once, whilst only 2% ($n = 3,798$) completed it on all five occasions. Table 1 also reports the mean, standard deviation (SD), and range for each nutrient. For example, the mean intake of iron was 13.50 mg (SD = 4.67 mg), with a maximum intake of 39.93 mg per day. Supplementary Fig. 2 provides a graphical representation of the distribution of each nutrient.

**Association of polygenic scores for psychiatric traits with nutrient intake**. In the mixed-effects regression analyses of the nutrients on polygenic scores of psychiatric disorders and traits, and behavioural and anthropometric traits, differential associations emerged. Figures 1 and 2 show the estimated $R^2$ values, scaled by the variance explained by the polygenic score predicting itself on the liability scale for Model 0 (i.e., baseline model) and Model 5 (i.e., full model) in configuration 1 (see Supplementary Figs. 3 and 4 for scaled plots, and Supplementary Figs. 5 and 6 for unscaled plots, for model configurations 1 and 2, respectively). We found no association between nutrient intake and polygenic scores for lupus (i.e., negative control, as expected), autism spectrum disorder, and food addiction.

*Schizophrenia and bipolar disorder*. Polygenic scores for schizophrenia and bipolar disorder had the highest number of

**Table 1 Descriptive statistics: age, sex, number of questionnaires completed, and nutrient intakes in 163,619 participants of the UK Biobank cohort.**

| Age | 56.22 (7.89) | 40−72 |
|---|---|---|
| | *n* (%) | |
| *Gender* | Males | 73,853 (45.1%) |
| | Females | 89,766 (54.9%) |
| | Total | 163,619 |
| *Number of questionnaires completed* | 1 | 65,296 (39.9%) |
| | 2 | 39,254 (24%) |
| | 3 | 33,539 (20.5 %) |
| | 4 | 21,732 (13.3%) |
| | 5 | 3,798 (2.3%) |
| *Nutrient intakes (units)* | Mean (SD) | Range |
| Alcohol (g) | 16.18 (22.73) | 0–149.63 |
| Protein (g) | 80.93 (29.96) | 0–294.54 |
| Carbohydrates (g) | 249.78 (87.66) | 0–834.88 |
| Fats (g) | 76.19 (31.97) | 0–270.35 |
| Fibre (g) | 16.32 (7.02) | 0–59.3 |
| Food weight (g) | 3162.84 (808.55) | 0–6361.5 |
| Folate (µg) | 298.17 (118.65) | 0–995.36 |
| Calcium (mg) | 961.25 (386.64) | 0–3934.24 |
| Carotene (µg) | 3044.85 (2860.47) | 0–24992.8 |
| Iron (mg) | 13.50 (4.67) | 0–39.93 |
| Vitamin B6 (µg) | 2.15 (0.78) | 0–6 |
| Vitamin B12 (µg) | 6.13 (4.73) | 0–49.33 |
| Vitamin C (µg) | 150.14 (110.63) | 0–997.45 |
| Vitamin D (µg) | 2.81 (3.22) | 0–24.76 |
| Vitamin E (µg) | 9.13 (4.77) | 0–47.31 |

*Note.* values are mean (standard deviation) and range (min−max) for age and nutrient intakes, and *n* (%) for sex and number of assessments completed.

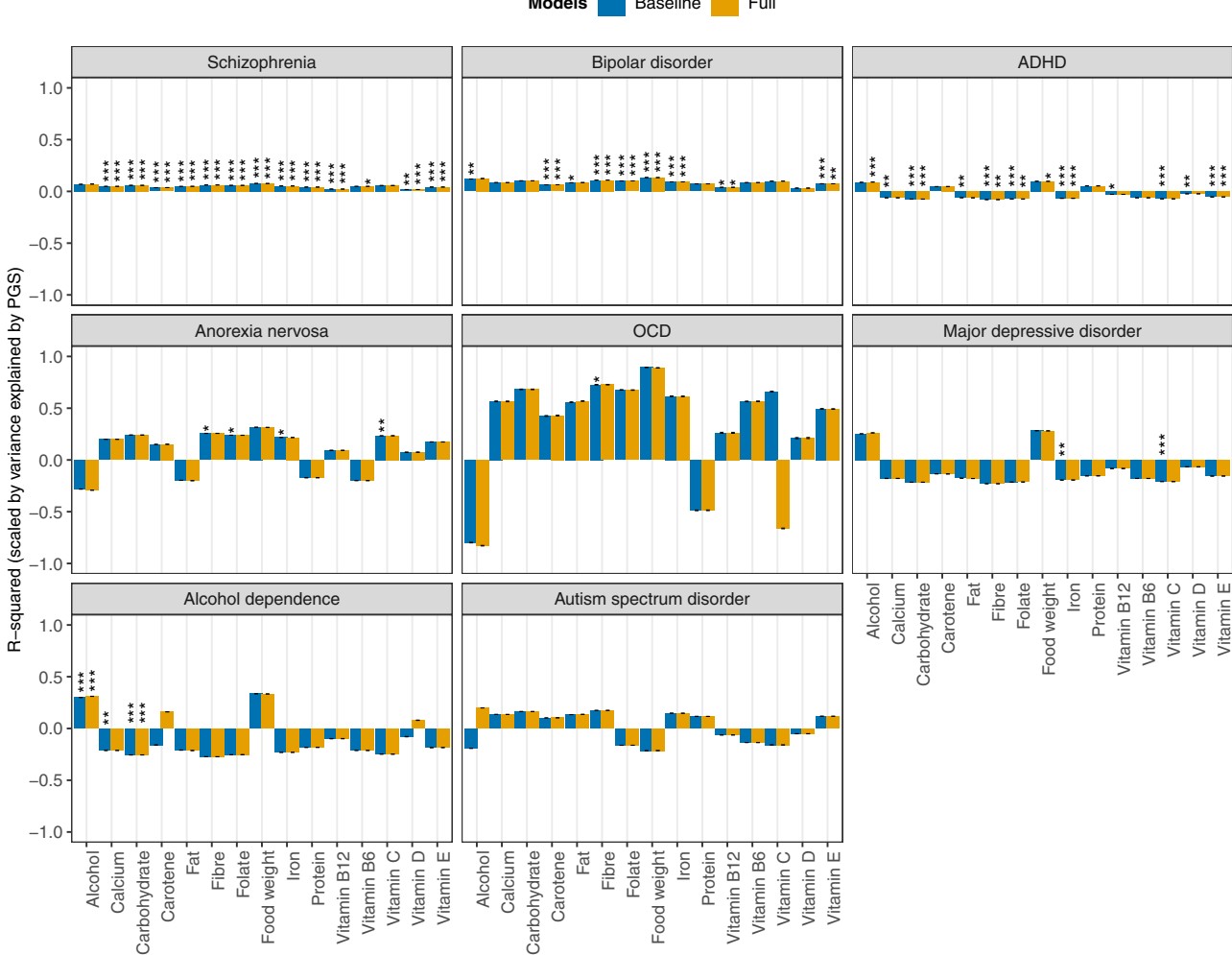

**Fig. 1 Associations between polygenic scores for psychiatric disorders and nutrient intake.** Results are shown from linear mixed-effects model analyses. $Y$-axis shows the $R^2$ estimates which have been scaled by the variance explained by the polygenic score predicting itself on the liability scale and have been multiplied by the direction of the coefficient estimate. Colours represent the different models: grey for Model 0 (i.e., baseline model) and yellow for Model 5 (i.e., full model) in configuration 1. Error bars represent standard errors and asterisks indicate statistically significant estimates. Bonferroni-corrected $p$-value thresholds: $* = p < 0.05/132$, $** = p < 0.01/132$, $*** = p < 0.001/132$. Data are represented for 163,619 participants of the UK Biobank cohort. The full results can be found in Supplementary Figs. 3 and 4.

associations with nutrient intake (Supplementary Fig. 5). The schizophrenia polygenic score was modestly but positively associated with most nutrients, with the exception of alcohol and vitamin C. We found a positive association between schizophrenia polygenic score and vitamin B6 but only in the full model (Model 5 in Supplementary Fig. 3/5 or Model 4 in Supplementary Fig. 4/6). A one SD higher bipolar disorder polygenic score was associated with higher food weight (11.4 g) and a higher intake of alcohol (0.20 g), fats (0.24 g), fibre (0.11 g), folate (1.45 μg), carotene (27.6 μg), iron (0.06 mg) and the vitamins B12 (0.03 μg) and E (0.05 μg) (Model 0 in Supplementary Data 1/2), as well as calcium (effect size after adjusting for ill-health = 2.94 mg), and the vitamins C (0.94 μg) and D (0.02 μg) after adjusting for either the typicality and kind of diet followed, physical activity or ill health (Supplementary Fig. 3/5 and Supplementary Data 1). After adjusting for phenotypic SES and EA, the associations between the bipolar disorder polygenic score and the intake of alcohol, fats, and vitamins C, D, and B12 attenuated and did not remain significant (Supplementary Fig. 3/5). We also observed no association between the bipolar disorder polygenic score and fats

when fixed effects were added in a stepwise method (Supplementary Fig. 4/6).

*Anorexia nervosa and OCD.* A one SD higher anorexia nervosa polygenic score was associated with a higher intake of fibre (0.06 g), folate (0.93 μg), iron (0.03 mg), and vitamin C (0.92 μg; Fig. 1; Model 0 in Supplementary Data 1/2). For fibre, folate, and iron, associations were not significant after adjusting for SES, EA, and physical activity, as well as ill health for iron—irrespective of modelling approach (Supplementary Figs. 3/5 and 4/6). A higher OCD polygenic score was associated with 0.06 g higher fibre intake (Model 0 in Supplementary Data 1/2). Again, this association attenuated after adjusting for phenotypic SES and EA, and did not remain significant. Interestingly, OCD showed the greatest effect size on nutrient intake, proportional to the power of the polygenic score (Fig. 1; Supplementary Figs. 3 and 4).

*MDD.* A higher polygenic score for MDD was associated with a lower intake of iron (−0.04 mg) and vitamin C (−1.13 μg; Fig. 1; Model 0 in Supplementary Data 1/2). We also observed a positive

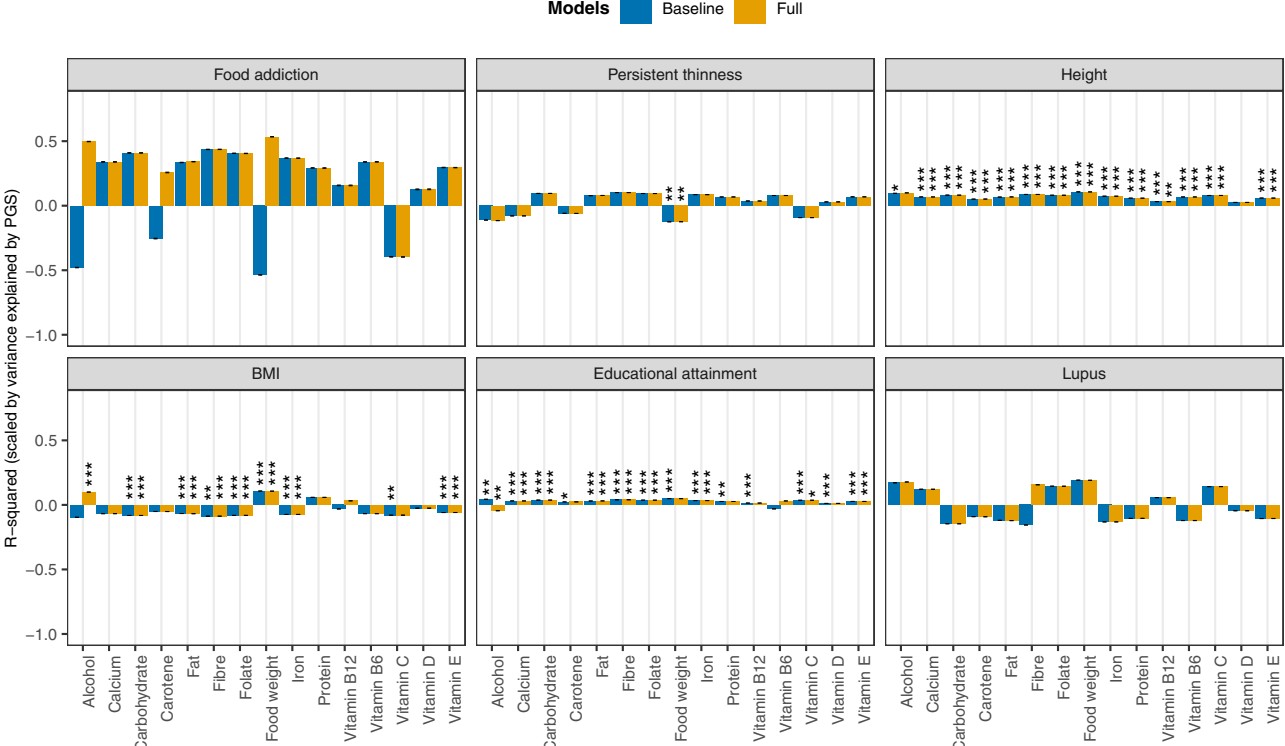

**Fig. 2 Associations between polygenic scores for psychiatric, behavioural, and anthropometric traits and nutrient intake.** Results are shown from linear mixed-effects model analyses. Y-axis shows the $R^2$ estimates which have been scaled by the variance explained by the polygenic score predicting itself on the liability scale and have been multiplied by the direction of the coefficient estimate. Colours represent the different models: grey for Model 0 (i.e., baseline model) and yellow for Model 5 (i.e., full model) in configuration 1. Error bars represent standard errors and asterisks indicate statistically significant estimates. Bonferroni-corrected p-value thresholds: $* = p < 0.05/132$, $** = p < 0.01/132$, $*** = p < 0.001/132$. Data are represented for 163,619 participants of the UK Biobank cohort. The full results can be found in Supplementary Figs. 3 and 4.

association with alcohol intake and a negative association with vitamin E after adjusting for physical activity. We found the association between MDD polygenic score and iron attenuated and did not remain significant when adjusting for SES and EA or ill health (Supplementary Figs. 3/5 and 4/6). In addition, stepwise inclusion of additional fixed effects attenuated the association between MDD polygenic score and vitamin C which also did not remain significant (Supplementary Fig. 3/5).

*ADHD and alcohol dependence.* A one SD higher ADHD polygenic score was associated with 0.23 g higher alcohol intake and a 7.6 g higher overall food weight (Fig. 1). These associations were not significant after adjusting for physical activity, as well as ill-health (Supplementary Fig. 3/5). However, they remained significant when fixed effects were adjusted for in a stepwise manner (Model 4 in Supplementary Fig. 4/6). A one SD higher ADHD polygenic score was also associated with a lower intake of carbohydrates (−1.36 g), fats (−0.29 g), fibre (−0.08 g), folate (−1.48 µg), calcium (−3.45 mg), iron (−0.09 mg), and the vitamins B12 (−0.03 µg), C (−1.26 µg), D (−0.02 µg), and E (−0.09 µg) (Model 0 in Supplementary Data 1/2). However, associations with fats, calcium, and the vitamins C and D were not significant after adjusting for SES and EA, as well as physical activity for fats and vitamin B12, and ill health for vitamin D—irrespective of whether fixed effects were grouped or added in a stepwise manner. Similarly to ADHD, a higher alcohol dependence polygenic score was associated with 0.4 g higher alcohol intake and 5.9 g higher food weight (Fig. 1; Model 0 in Supplementary Data 1/2), the latter only when adjusting for SES and EA. The association between alcohol dependence polygenic score and alcohol intake

was dependent on SES and EA, and reduced when adjusting for alcohol consumption (as expected, Supplementary Figs. 3/5 and 4/6). We also found negative associations with the intake of carbohydrates (−1.18 g) and calcium (−3.42 mg) (Model 0 in Supplementary Data 1/2). The latter was not significant after adjusting for ill health.

*Height, BMI, and persistent thinness.* We also studied polygenic scores for height, BMI, and persistent thinness (Fig. 2). As with schizophrenia, the height polygenic score was positively associated with most nutrients, except vitamin D. After adjusting for phenotypic SES and EA the associations between the height polygenic score and alcohol intake attenuated and did not remain significant (Supplementary Figs. 3/5 and 4/6). In addition, a one SD higher BMI polygenic score was associated with 0.2 g higher alcohol intake and 9.67 g higher food weight, but lower intake of carbohydrates (−1.90 g), fats (−0.58 g), and fibre (−0.07 g) (Model 0 in Supplementary Data 1/2). The association between the BMI polygenic score and alcohol intake was restricted to those models adjusting for ill health (i.e., Models 4 and 5 in Supplementary Fig. 3/5 or Models 3 and 4 in Supplementary Fig. 4/6). As a sensitivity analysis, we report similar findings for a body fat percentage polygenic score (Supplementary Fig. 7). Furthermore, we found a one SD higher polygenic score for persistent thinness was associated with 8.61 g lower food weight (Model 0 in Supplementary Data 1/2).

*Educational attainment.* Finally, we found a one SD higher polygenic score for EA was positively associated with most nutrients, except vitamin B6 (Fig. 2). Adjusting for ill-health

reversed the association between the EA polygenic score and alcohol intake (0.20 g in Model 0 compared to −0.28 g in Model 4 ill-health adjustment—Supplementary Data 1). In addition, we found adjustment for phenotypic SES and EA weakened associations, with protein, food weight, carotene, and vitamin B12 not meeting our significance threshold anymore (Supplementary Fig. 3/5).

## Discussion

A recent UK Biobank study found significant genetic correlations between schizophrenia and two diet groups, one representing a meat-related diet and the other a fish and plant-related diet[19]. This spurred our in-depth investigation of the association between polygenic scores for psychiatric disorders and nutrient intake. We found that polygenic scores for seven psychiatric disorders and several behavioural and anthropometric traits are significantly associated with self-reported nutrient intake on an average day.

Polygenic scores for schizophrenia, bipolar disorder, and ADHD showed the highest number of significant associations with the intake of specific nutrients. However, scaling the calculated $R^2$ values by their power (the variance explained by the polygenic score predicting itself on the liability scale) revealed that the OCD polygenic score had the greatest effect sizes on nutrient intake. Given that the OCD GWAS was relatively small compared to the other psychiatric disorder GWAS, these findings suggest that with larger and more powerful OCD GWAS associations of OCD with diet traits may become more apparent.

We also investigated polygenic scores for anorexia nervosa and persistent thinness. Both are characterised by low BMI but differ in their psychiatric symptoms[21]. Individuals with persistent thinness do not suffer from undernutrition or exhibit any typical clinical features, such as amenorrhea, fear of weight gain, or hormonal abnormalities, commonly seen in anorexia nervosa[22]. Notably, polygenic scores for anorexia nervosa and persistent thinness had distinct associations with nutrient intake. The anorexia nervosa polygenic score was associated with higher intake of fibre, folate, iron, and vitamin C. In contrast, a higher persistent thinness polygenic score was associated with lower food weight but nothing else. These findings complement the clinical definition of persistent thinness and support the suggestion that anorexia nervosa and persistent thinness may be genetically distinct[23].

Polygenic scores for ADHD and alcohol dependence had a similar pattern of association with nutrient intake, both being associated with higher alcohol intake. Given that impulsivity is common among individuals with ADHD and those who are alcohol dependent[24,25], alcohol use may relate to an individual's impulsive tendency to use alcohol for the immediate reward associated with drinking[26]. The association between genetic risk for alcohol dependence and alcohol intake was not significantly influenced by phenotypic SES and EA. This is interesting because SES and EA have been identified as prominent risk factors for alcohol dependence[27,28]. However, our findings suggest that other unaccounted risk factors, such as peer pressure[29] and living in a family or culture where frequent alcohol use/abuse is accepted[30], may be more influential in an individual's genetic liability to develop an alcohol problem.

A higher EA polygenic score was positively associated with nutrient intake, providing a genetic basis for the phenotypic association between EA and dietary intake[31]. Adjusted for phenotypic EA attenuated associations, as expected. However, several remained significant including associations with carbohydrate and fat intake. This could reflect genetic pleiotropy as a recent study found that the EA polygenic score captures DNA variants shared between educational achievement and personality traits, including agreeableness, openness, conscientiousness, and academic motivation[32]. Personality traits have previously been linked to taste preference and eating

behaviour[33]. This, therefore, offers a potential explanation for the significant associations between EA polygenic score and specific nutrients after adjusting for phenotypic EA.

Contrary to our expectations, a higher BMI polygenic score was associated with lower intake of carbohydrates, fats, and fibre. We observed similar findings for a body fat percentage polygenic score. This suggests that higher BMI in obese individuals may not originate from a biological liability for higher fat and carbohydrate intake but may be associated with other factors such as dysfunction of autonomic neural circuits[34]. Alternatively, underreporting bias may have occurred because individuals with a higher BMI polygenic score reported higher food intake, when measured as weight, but lower intake of specific nutrients. This misreporting may reflect socially desirable responses and low ability to report own dietary intake[35]. Based on our findings, these socially desirable responses may be a lower reported intake of fats, carbohydrates, and total energy intake than actual intake.

Finally, we grouped fixed effects into distinct groups to determine which environmental factors influence the association between polygenic scores for psychiatric disorders and nutrient intake. Using this approach, we found some evidence for collider bias whereby the exposure and outcome independently cause a third variable, inducing associations where there is no true effect. For example, no association was observed between the alcohol dependence polygenic score and food weight until we adjusted for SES and EA. Given that food consumption is socially stratified[36–38], our observation suggests that genetic liability to alcohol dependence may partially drive the association between SES and dietary behaviour. Furthermore, this suggests that environmental factors commonly believed to confound the relationship between psychiatric disorders and dietary behaviour may act as colliders. Future studies should identify potential colliders and their magnitude of effect.

There are several important caveats that need to be taken into account when interpreting these findings. First, dietary intake in the UK Biobank was self-reported, as with most nutritional epidemiology studies, and this method of data collection has inherent limitations[39,40]. We used repeated measures to reduce reporting bias but objective measures of dietary intake, which are currently unavailable at these large sample sizes, would be superior. Second, the observed absolute effect sizes were small (e.g one SD higher ADHD polygenic score was associated with 0.11 g higher alcohol intake and −0.28 g lower fat intake) and, therefore, we caution against overinterpreting the associations between polygenic scores and the intake of specific nutrients. Furthermore, although our polygenic scores were constructed from the largest available GWAS, some traits still had relatively small sample sizes and consequently, these polygenic scores were more weak predictors. We attempted to address this by scaling polygenic scores however these analyses should be repeated when sample sizes have increased.

To summarise, polygenic propensity for a psychiatric disorder is associated with nutrient intake. This has important implications for future treatment strategies. Our findings encourage further research into the shared biological pathways and common environmental factors influencing psychiatric disorders and nutrient intake. This could help develop integrative treatments that prevent the development of additional comorbidities in individuals with a psychiatric diagnosis. Future research should explore the developmental association between psychiatric disorders and nutrient intake, to capture age-dependent differences. These studies should focus on the impact of SES, EA, and physical activity because these factors influenced several associations observed. In addition, we found having genetic risk for schizophrenia was associated with higher fat intake and food weight even after controlling for antipsychotics. However, Mendelian

randomisation using GWAS findings suggests schizophrenia is negatively associated with body composition[41]. Future work should attempt to identify potential causes underlying these differential associations. Apps focused on health and fitness have emerged on the smartphone market: these studies should take advantage of food-tracking apps which may provide a better alternative to dietary recall questionnaires.

## Methods

**Ethics**. We obtained approval for this research under an approved access request (application 23395) to UK Biobank. UK Biobank has approval from the North West Multi-centre Research Ethics Committee, which covers the UK, and the Patient Information Advisory Group for gaining access to information that would allow it to invite people to participate. Our use of the data was governed by the analysis plan in our access request and the terms of the material transfer agreement between King's College London and UK Biobank. We assert that all procedures contributing to this work comply with the ethical standards of the relevant national and institutional committees on human experimentation and with the Helsinki Declaration of 1975, as revised in 2008.

**Study population, genotype quality control, and sample size**. The UK Biobank is a large prospective cohort study consisting of approximately 500,000 participants aged 46−69 years when recruited in 2006−2010[42]. Written informed consent was obtained from all participants. The study assessed dietary behaviour using a web-based 24-h dietary assessment tool, which asked about the frequency of consumption of common foods and drinks (Category 100090)[43]. Participants were asked whether what they ate and drank yesterday was typical (Data-Field 100020) and if they routinely followed a special diet (Data-Field 20086). Responses were automatically coded to provide estimated daily nutrient intake (Category 100098). Participants completed the initial assessment at recruitment centres and then remotely on four occasions between April 2009 and June 2012 (see Supplementary Methods for further information).

Genome-wide genetic data came from the full release of the UK Biobank ($n =$ 487,410) and were processed according to the quality control pipeline[44]. Standard genotype quality control criteria were used[45]. Thresholds were variants with a minor allele frequency > 1%, directly genotyped or imputed (IMPUTE INFO metric > 0.4[46]). For individuals, a genotype call rate > 98%, concordant phenotypic and genetic gender information, removing third-degree relatives and closer[47]. Our analyses were limited to individuals of European ancestry due to insufficient numbers of other ancestry groups.

Pregnant females were removed as well as potential outliers in the dataset using lower and upper cutoff limits to each estimated nutrient. These cutoffs were identified by generating scatterplots. We included all time points each participant answered the questionnaire, but restricted our analyses to individuals with complete data on variables that may potentially influence nutrient intake, such as socioeconomic status (SES) and educational attainment (EA). Table 2 documents the exclusion of UK Biobank participants. Our sample after exclusion consisted of 350,339 data entries for each estimated nutrient intake covering 163,619 participants (i.e., 77.5% of the original sample). Data cleaning was performed in R version 3.5.3.

**Table 2 Exclusion of UK Biobank participants.**

**Number of participants who completed the Diet by 24-h recall questionnaire at least once = 211,039**

| Exclusion criteria | Number of participants excluded |
| --- | --- |
| No genotype data | 43,036 |
| Pregnant (included those that were doubtful) | 132 |
| Missing data on variables that may potentially influence nutrient intake<br>- Age<br>- Sex<br>- SES<br>- Educational attainment<br>- Physical activity<br>- Smoking<br>- Alcohol consumption<br>- Diagnoses and medication that affects dietary intake | 4,252 |
| Participants remaining | 163,619 |

**Nutritional intake data**. Nutrient intake (Category 100098) was pre-calculated by the UK Biobank (see Supplementary Methods for further information). We excluded one of each pair of nutrients with correlations >0.7 (Supplementary Fig. 1). Accordingly, energy, total sugar, starch, saturated and polyunsaturated fats, magnesium, potassium, and retinol were not analysed. These estimates were excluded, except magnesium, due to their high correlation with the key macronutrients: protein, carbohydrates, and fats. There was a high correlation between iron and magnesium intake. We kept iron because iron deficiency is the most prevalent nutritional deficiency and a potential risk factor of psychiatric disorders[6]. In total, we studied the intake of 14 nutrients—protein, carbohydrates, fats, fibre, food weight, folate, calcium, carotene, iron, and vitamins B12, B6, C, D, and E— and alcohol.

**Polygenic scores**. Polygenic scores for psychiatric disorders were constructed for each UK Biobank participant using PRSice version 2.2.1 (plink-clump-p 1-clump-r2 0.1-clump-kb 250-perm 10000)[48]. Single-nucleotide polymorphism (SNP) weights were based on the output from GWAS of each trait excluding UK Biobank participants (Table 3). We also investigated polygenic scores for BMI, body fat percentage, height, and persistent thinness as body composition is associated with nutrient intake. The latter to compare low BMI with anorexia nervosa—both have low BMI in common, but they differ in their psychiatric symptoms[21]. We included polygenic scores for educational attainment, because it has a negative genetic correlation with body composition[41], and lupus as a negative control, because we expected it not to be associated with nutrient intake[49,50]. Finally, a polygenic score for food addiction was included to investigate what components of food intake may promote an addictive-like response in individuals. To minimise multiple testing, we selected SNPs with $P$-values <0.2 into the scores.

**Linear mixed-effects models**. We adopted a linear mixed-effects modelling approach to determine whether having an underlying genomic risk for a psychiatric disorder influences nutrient intake while accounting for correlations among repeated assessments within an individual[51]. Specifically, we used the lmerTest package in R[52] which extends the 'lmerMod' class of the lme4 package by providing $P$-values for tests for fixed effects. We also used the 'MuMIn' package[53] which calculates $R$-squared values for mixed-effects models.

*Main analyses.* Our baseline mixed-effects model (Model 0; Table 4) included the following fixed effects: polygenic score studied, age, sex, and the first six ancestry principal components (PCs) calculated on the European subsample. The number of time points each participant answered the dietary assessment was also included as a random effect to account for repeated measures. Table 4 summarises the additional fixed effects included in each of the subsequent models. To identify environmental factors having an important influence on associations between polygenic scores for psychiatric disorders and nutrient intake, additional fixed effects were grouped and differently assessed in each model—Model 1) typicality and kind of diet followed, Model 2) SES and EA, Model 3) physical activity, and Model 4) ill-health, including negative health behaviours and appetite-modulating medication—with Model 5 adjusting for all fixed effects (configuration 1 in Table 4).

To aid in the interpretation of estimated effect sizes, given varying discovery sample sizes for their derivation, polygenic scores were scaled for graphical presentation by dividing the calculated $R^2$ by the variance explained by the ability of each polygenic score to predict itself on the liability scale. This allowed us to determine the relative importance of each disorder and trait for each nutrient intake, given that some disorders and traits (e.g., schizophrenia, height) had more powerful polygenic scores available. We obtained bootstrapped standard errors (SE) for the $R^2$ statistics using the 'boot' package in R, with 100 replications.

*Supplementary/sensitivity analyses.* We tested for potential collider bias by including Model 0 (Table 4) to see if an association between the exposure (polygenic score) and outcome (nutrient) existed before adding additional fixed effects. In addition, we present an alternative set of models whereby fixed effects were added in a stepwise manner as opposed to being differentially assessed (Configuration 2 in Table 4), to see whether patterns of association between polygenic scores for psychiatric disorders and nutrient intake are affected by different approaches to add fixed effects.

**Multiple comparisons**. Multiple testing correction was performed using matrix decomposition of the correlation matrix of all traits studied (anorexia nervosa, ADHD, OCD, schizophrenia, MDD, alcohol dependence, persistent thinness, food addiction, height, educational attainment, lupus, alcohol, protein, carbohydrates, fats, fibre, food weight, folate, calcium, carotene, iron, and vitamins B12, B6, C, D, and E) to identify the number of independent tests to adjust the $p$-value threshold using Bonferroni correction (not considering supplementary analyses; see Supplementary Methods for further information). $p < 0.05/132$ was considered as statistically significant.

**Table 3 Table summarising genome-wide association study (GWAS) discovery sample size, SNP-based heritability on the observed scale.**

| Trait | GWAS sample | Observed scale SNP-based heritability |
|---|---|---|
| Anorexia nervosa[54] | 16,224 cases and 52,460 controls | 17.4% (1.2%) |
| Obsessive–compulsive disorder[55] | 2,688 cases and 7,037 controls | 33.8% (4.8%) |
| Educational attainment[56] | 766,345 individuals | 10.7% (0.2%) |
| Schizophrenia[9] | 33,610 cases and 43,456 controls | 45.5% (1.6%) |
| Attention-deficit/hyperactivity disorder[57] | 19,099 cases, 34,194 controls | 24% (1.5%) |
| Alcohol dependence[58] | 11,569 cases and 34,999 controls | 5.4% (1%) |
| Major depressive disorder[59] | 116,404 cases and 314,990 controls | 5.7% (0.2%) |
| Bipolar disorder[60] | 20,352 cases and 31,358 controls | 21% (1.1%) |
| Autism spectrum disorder[61] | 18,381 cases and 27,969 controls | 12% (1.0%) |
| Food addiction[62] | 9,314 females | 10% (4%) |
| Persistent thinness[23] | 1,471 cases and 6,460 controls | 16.7% (3.9%) |
| Height[63] | 253,288 individuals | 31.2% (1.4%) |
| Body mass index[64] | 322,154 individuals | 13% (0.5%) |
| Body fat percentage[65] | 100,716 individuals | 10.4% (0%) |
| Systemic lupus erythematosus[66] | 7,219 cases and 15,991 controls | 33.3% (9.7%) |

**Table 4 Table summarising the fixed effects included in the linear mixed-effects models. Configurations 1 and 2 represent the models presented in Supplementary Figs. 3/5 and 4/6, respectively.**

| Configuration 1 - fixed effects grouped and differently assessed | | |
|---|---|---|
| Model | Analysis | Fixed effects |
| 0 | Supplementary | Polygenic score under investigation, sex, age + principal components 1—6 |
| 1 | Main | Model 0 + special diet and typical diet yesterday |
| 2 | Main | Model 1 + socioeconomic status and educational attainment |
| 3 | Main | Model 1 + physical activity |
| 4 | Main | Model 1 + smoking, alcohol consumption and diagnoses and medication that affect dietary intake |
| 5 | Main | All fixed effects |
| Configuration 2 - fixed effects grouped and added in a stepwise manner | | |
| Model | Analysis | Fixed effects |
| 0 | Supplementary | Polygenic score under investigation, sex, age + principal components 1—6 |
| 1 | Supplementary | Model 0 + special diet and typical diet yesterday |
| 2 | Supplementary | Model 1 + socioeconomic status and educational attainment |
| 3 | Supplementary | Model 2 + smoking, alcohol consumption and diagnoses and medication that affect dietary intake |
| 4 | Supplementary | Model 3 + physical activity |

**Reporting summary**. Further information on research design is available in the Nature Research Reporting Summary linked to this article.

## Data availability

Authors had full access to the data supporting the findings of this study. UK Biobank is an open access resource. Data are available to bona fide scientists, undertaking health-related research that is in the public good. All individual-level data from UK Biobank can be accessed by applying to the UK Biobank Central Access Committee (http://www.ukbiobank.ac.uk/register-apply/). Source data are provided in Supplementary Data 1/2. The datasets used in this study are available from A.K.H. on reasonable request.

## Code availability

Analysis code can be accessed on https://github.com/AvinaHunjan/scripts. Software can be accessed for PRSice, at https://choishingwan.github.io/PRSice/.

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

## Acknowledgements

This research has been conducted using the UK Biobank Resource under application number 23395 (with thanks to C.H.). We are extremely grateful to all the participants who took part in this study, and the whole UK Biobank team. This study represents independent research partially funded by the National Institute for Health Research (NIHR) Biomedical Research Centre at the South London and Maudsley National Health Service (NHS) Foundation Trust and by King's College London. The views expressed are those of the authors and not necessarily those of the NHS, the NIHR, or the Department of Health and Social Care. High-performance computing facilities were funded with capital equipment grants from the Guy's and St. Thomas' Charity (TR130505) and Maudsley Charity (980). T.C.E. and G.B. were partially funded by a programme grant from the UK Medical Research Council (MR/M021475/1). C.H. acknowledges funding from Lundbeckfonden (R276-2018-4581).

## Author contributions

All authors substantially contributed to the design of the research. A.K.H., C.H., and Y.L. provided essential materials, analysed data, or performed statistical analyses. A.K.H.

wrote the paper, and all authors drafted the work, approved the final paper, and agreed to be accountable for all aspects of the work. T.C.E. and G.B. had primary responsibility for the final content.

## Competing interests
G.B. has received grant funding from and served as a consultant to Eli Lilly, has received honoraria from Illumina, and has served on advisory boards for Otsuka and Compass Pathways. All other authors have no competing interests.
