## [Transparent Peer Review File · Communications Biology]

Reviewers' comments:

Reviewer #1 (Remarks to the Author):

This is extremely well-written manuscript containing substantial analysis exploring the associations between nutritional intake and psychiatric disorders, psychiatric-related traits, body composition measurements, and educational attainment alongside lupus as a negative control. The findings from this work are important for understanding the biological relationship between the described traits and nutrition. I do have the following suggestions to help further improve the manuscript:

- 1) A figure documenting the exclusion of UKB participants from the final cohort would be informative.
- 2) I would prefer for the decision to select lupus as a negative control to be evidence-based – i.e. appropriate reference to be provided.
- 3) The observed absolute changes in nutrient intake are often very small. The authors need to cite this in the discussion/limitations and provide relevant context for practitioners.

Minor

- 1) typo Introduction: "In addition, vitamin deficiencies is" to In addition, vitamin deficiencies are
- 2) Ethics subsection: redundant acronyms MREC and PIAG
- 3) Height, BMI and persistent thinness subsection: "0.2 mg higher alcohol" – should this be g rather than mg?

Reviewer #2 (Remarks to the Author):

I was asked to review the paper "Association between polygenic propensity for a psychiatric disorder and nutrient intake". The paper is a solid research and necessary research question, sure to influence the psychiatric-nutrition literature. The methods are well-done, but there are areas of reporting could be improved. I have further comments below.

I suggest the authors change the end of the first paragraph, PRS are not causal models either and tell us little about potential causality. Some of the author have previously published on MR methods, so I am confused as to why those publications are not mentioned?

When choosing highly correlated outcomes, how exactly did the authors choose 1 of each pair to exclude? Which one of the pair specifically? Also .7, is a bit low. I'm curious if the matrix decomposition was spectral decomposition? If so, I think the authors shouldn't be removing traits unless the r is .9, and even then spectral decomposition should still handle the model effectively. I think it would be better to test all traits and correct for all fairly.

Were the polygenic scores pruned? A sentence could be spared on what exactly the parameters in PRSice were.

.2 seems like an arbitrary threshold to train the PRS. I don't quite understand why this threshold was used and not another that is more standard like .1 (trending) or .05? I get these choices are arbitrary but it is a weakness of PRSice. Alternative methods (like PRS-cs auto) would not force the authors to make that choice, so the justification for PRSice over other methods is needed. I will let the authors know that food addiction is a small enough sample that more advanced PRS methods won't work, but it's still worth noting that these methods are out there at least, and perhaps even using them for the phenotypes they are well-powered for (schizophrenia for example).

I like that the authors avoided collider bias by presenting models with and without the specific social science covariates.

Saying scores are attenuated is a bit vague. I think telling us the r^2 change and whether it remained significant is necessary. Perhaps attenuated down means that it remained significant? It would be worth saying that.

Discussion on collider bias is interesting, but perhaps needs an extra sentence? Why can we draw when the alcohol prs is only predictive when controlling for SES? What does this mean for nutrition? Which has a very socially stratified variability.

We thank the reviewers for their careful and considered advice. We have addressed their queries and suggest, making changes throughout the text. We believe the manuscript is now considerably improved because of this.

We itemise the reviewers’ comments below and our responses. We have mirrored and tracked these changes in the main manuscript file also.

REMARKS TO AUTHOR:

Reviewer: 1

This is extremely well-written manuscript containing substantial analysis exploring the associations between nutritional intake and psychiatric disorders, psychiatric-related traits, body composition measurements, and educational attainment alongside lupus as a negative control. The findings from this work are important for understanding the biological relationship between the described traits and nutrition. I do have the following suggestions to help further improve the manuscript:

R1P1 A figure documenting the exclusion of UKB participants from the final cohort would be informative.

We thank the reviewer for this great idea. We have now provided the table below on page 6 of the manuscript

Table 1: Exclusion of UK Biobank participants

Number of participants who completed the Diet by 24-hour recall questionnaire at least once = 211,039	
Exclusion criteria	Number of participants excluded
No genotype data	43,036
Pregnant (included those that were doubtful)	132
Missing data on variables that may potentially influence nutrient intake  - Age - Sex - SES - Educational attainment - Physical activity - Smoking - Alcohol consumption - Diagnoses and medication that affect dietary intake 	4,252
Participants remaining	163,619

We also state in text, “Table 1 documents the exclusion of UK Biobank participants” Page 6.

R1P2 I would prefer for the decision to select lupus as a negative control to be evidence-based – i.e. appropriate reference to be provided.

We thank the reviewer for this comment. We have provided the following references:

- Costenbader K.H., Kang J.H., Karlson E.W. Antioxidant intake and risks of rheumatoid arthritis and systemic lupus erythematosus in women. *Am. J. Epidemiol.* 2010;172:205–216. doi: 10.1093/aje/kwq089
- Hiraki LT, Munger KL, Costenbader KH, Karlson EW. Dietary intake of vitamin D during adolescence and risk of adult-onset systemic lupus erythematosus and rheumatoid arthritis. *Arthritis Care Res.* (2012) 64:1829–36. 10.1002/acr.2177

These studies show that definitive evidence is lacking that dietary factors influence human SLE disease development or disease activity. Costenbade et al. found that the intake of vitamins A, C, and E and α -carotene, β -carotene, β -cryptoxanthin, lycopene, lutein, and zeaxanthin from food is not associated with the risk of developing systemic lupus erythematosus. Hiraki et al. reported similar findings for vitamin D.

R1P3 The observed absolute changes in nutrient intake are often very small. The authors need to cite this in the discussion/limitations and provide relevant context for practitioners.

Thank you - this point is important. We have added the following sentence to the limitations section:

“Second, the observed absolute effect sizes were small (e.g. one SD higher ADHD polygenic score was associated with 0.11 g higher alcohol intake and -0.28 g lower fat intake) and, therefore, we caution against overinterpreting the associations between polygenic scores and the intake of specific nutrients”.
Page 19.

R1M1 typo Introduction: “In addition, vitamin deficiencies is” to In addition, vitamin deficiencies are

We thank the reviewer for noticing this typo. We have corrected it.

R2M2 Ethics subsection: redundant acronyms MREC and PIAG

We thank the reviewer for noticing these redundant acronyms. They have now been removed from our ethics subsection.

R3M3 Height, BMI and persistent thinness subsection: “0.2 mg higher alcohol” – should this be g rather than mg?

Again, we thank the reviewer for noticing this typo. We have corrected it.

Reviewer #2:

I was asked to review the paper “Association between polygenic propensity for a psychiatric disorder and nutrient intake”. The paper is a solid research and necessary research question, sure to influence the psychiatric-nutrition literature. The methods are well-done, but there are areas of reporting could be improved. I have further comments below.

R2P1 I suggest the authors change the end of the first paragraph, PRS are not causal models either and tell us little about potential causality. Some of the author have previously published on MR methods, so I am confused as to why those publications are not mentioned?

Some of the authors have indeed previously published on MR methods. However, these publications have not been cited in the manuscript due to not focusing on dietary intake but rather the association between psychiatric traits and body composition.

We have changed the end of the first paragraph from:

“However, these observations merely demonstrate associations and to establish cause and effect relationships between the development and/or severity of psychiatric disorders and alterations in dietary behavior, more in-depth investigation of their co-occurrence is needed.”

to:

“Taken together, these observations demonstrate the need for more in-depth investigation of the co-occurrence between psychiatric disorders and alterations in dietary behavior”. Page 3

R2P2 When choosing highly correlated outcomes, how exactly did the authors choose 1 of each pair to exclude? Which one of the pair specifically? Also .7, is a bit low. I’m curious if the matrix decomposition was spectral decomposition? If so, I think the authors shouldn’t be removing traits unless the r is .9, and even then spectral decomposition should still handle the model effectively. I think it would be better to test all traits and correct for all fairly.

We thank the reviewer for this comment. We chose .7 because i) we wanted to minimise multicollinearity and ii) our matrix was not sparse (few correlations greater than .9) containing a large number of pairs with correlations greater than .7. When it came to choosing 1 of each pair to exclude, we wanted to keep the three key macronutrients (carbohydrates, proteins and fat) to enable useful dietary recommendations. We found most nutrients we excluded were highly correlated with one of the three macronutrients. The only exception was the high correlation between dietary intake iron and magnesium. We choose to retain iron because “iron deficiency is the most prevalent nutritional deficiency and a potential risk factor of psychiatric disorders” Page 6.

R2P3 Were the polygenic scores pruned? A sentence could be spared on what exactly the parameters in PRSice were.

We used clumping rather than pruning. We have extended the following sentence “Polygenic scores for psychiatric disorders were constructed for each UK Biobank participant using PRSice version 2.2.1 (Choi and O’Reilly 2019)” in the methods to:

“Polygenic scores for psychiatric disorders were constructed for each UK Biobank participant using PRSice version 2.2.1 (plink --clump-p 1 --clump-r2 0.1 --clump-kb 250 --perm 10000; (Choi and O’Reilly 2019))”. Page 7.

R2P4 .2 seems like an arbitrary threshold to train the PRS. I don’t quite understand why this threshold was used and not another that is more standard like .1 (trending) or .05? I get these choices are arbitrary but it is a weakness of PRSice. Alternative methods (like PRS-cs auto) would not force the authors to make that choice, so the justification for PRSice over other methods is needed. I will let the authors know that food addiction is a small enough sample that more advanced PRS methods won’t work, but its still worth noting that these methods are out there at least, and perhaps even using them for the phenotypes they are well-powered for (schizophrenia for example).

We thank the reviewer for this comment. Like PRS-cs auto, PRSice does not require authors to provide an arbitrary threshold to train polygenic scores. However, we choose to train our polygenic scores at 0.2 because we have generally found that psychiatric disorder polygenic prediction maximises around 0.2 (e.g. see (Wray et al. 2018; Watson et al. 2019; Grove et al. 2019)), and wanted to enable as fair a comparison across all traits.

R2P5 Saying scores are attenuated is a bit vague. I think telling us the r^2 change and whether It remained significant is necessary. Perhaps attenuated down means that it remained significant? It would be worth saying that.

We thank the reviewer for this comment and have made it more clear which r^2 changes remained significant in the results section.

We replaced the sentence “*Similarly, a higher OCD polygenic score was associated with 0.06g higher fibre intake (Model 0 in Supplementary Table 1/2), attenuating when adjusted for EA and SES*” with “*A higher OCD polygenic score was associated with 0.06g higher fibre intake (Model 0 in Supplementary Table 1/2). Again, after adjusting for phenotypic SES and EA, this association attenuated and did not remain significant.*” Page 13.

The sentences “*We found the association between MDD polygenic score and iron attenuated when adjusting for SES and EA or ill health (Supplementary Figures 3/5 and 3/5). In addition, stepwise inclusion of additional fixed effects attenuated the association between MDD polygenic score and vitamin C (Supplementary Figure 3/5)*” has been edited to “*We found the association between MDD polygenic score and iron attenuated and did not remain significant when adjusting for SES and EA or ill health (Supplementary Figures 3/5 and 3/5). In addition, stepwise inclusion of additional fixed effects attenuated*

the association between MDD polygenic score and vitamin C which also did not remain significant (Supplementary Figure 3/5)” Page 13.

Lastly, we replaced the sentence “*The association between the height polygenic score and alcohol was attenuated by adjusting for SES and EA*” with “*After adjusting for phenotypic SES and EA the associations between the height polygenic score and alcohol attenuated and did not remain significant*” Page 14.

R2P6 Discussion on collider bias is interesting, but perhaps needs an extra sentence? Why can we draw when the alcohol prs is only predictive when controlling for SES? What does this mean for nutrition? Which has a very socially stratified variability.

We thank the reviewer for this comment. We have added an extra sentence as advised.

“Finally, we grouped fixed effects into distinct groups to determine which environmental factors influence the association between polygenic scores for psychiatric disorders and nutrient intake. Using this approach, we found some evidence for collider bias whereby the exposure and outcome independently cause a third variable, inducing associations where there is no true effect. For example, no association was observed between alcohol dependence polygenic score and food weight until we adjusted for SES and EA. Given that food consumption is socially stratified (Pechey et al. 2013; Appelhans et al. 2012; Darmon and Drewnowski 2008), our observation suggests that genetic liability to alcohol dependence may partially drive the association between SES and dietary behaviour. Furthermore, this suggests that environmental factors commonly believed to confound the relationship between psychiatric disorders and dietary behaviour may act as colliders. Future studies should identify potential colliders and their magnitude of effect.”

References

- Appelhans, Bradley M., Brandy-Joe Milliron, Kathleen Woolf, Tricia J. Johnson, Sherry L. Pagoto, Kristin L. Schneider, Matthew C. Whited, and Jennifer C. Ventrelle. 2012. “Socioeconomic Status, Energy Cost, and Nutrient Content of Supermarket Food Purchases.” *American Journal of Preventive Medicine* 42 (4): 398–402.
- Choi, Shing Wan, and Paul O’Reilly. 2019. “SA20 - PRSice 2: POLYGENIC RISK SCORE SOFTWARE (UPDATED) AND ITS APPLICATION TO CROSS-TRAIT ANALYSES.” *European Neuropsychopharmacology: The Journal of the European College of Neuropsychopharmacology* 29 (January): S832.
- Darmon, Nicole, and Adam Drewnowski. 2008. “Does Social Class Predict Diet Quality?” *The American*

- Journal of Clinical Nutrition* 87 (5): 1107–17.
- Grove, Jakob, Stephan Ripke, Thomas D. Als, Manuel Mattheisen, Raymond K. Walters, Hyejung Won, Jonatan Pallesen, et al. 2019. “Identification of Common Genetic Risk Variants for Autism Spectrum Disorder.” *Nature Genetics* 51 (3): 431–44.
- Pechey, Rachel, Susan A. Jebb, Michael P. Kelly, Eva Almiron-Roig, Susana Conde, Ryota Nakamura, Ian Shemilt, Marc Suhrcke, and Theresa M. Marteau. 2013. “Socioeconomic Differences in Purchases of More vs. Less Healthy Foods and Beverages: Analysis of over 25,000 British Households in 2010.” *Social Science & Medicine* 92 (September): 22–26.
- Watson, Hunna J., Zeynep Yilmaz, Laura M. Thornton, Christopher Hübel, Jonathan R. I. Coleman, Hélène A. Gaspar, Julien Bryois, et al. 2019. “Genome-Wide Association Study Identifies Eight Risk Loci and Implicates Metabo-Psychiatric Origins for Anorexia Nervosa.” *Nature Genetics* 51 (8): 1207–14.
- Wray, Naomi R., Stephan Ripke, Manuel Mattheisen, Maciej Trzaskowski, Enda M. Byrne, Abdel Abdellaoui, Mark J. Adams, et al. 2018. “Genome-Wide Association Analyses Identify 44 Risk Variants and Refine the Genetic Architecture of Major Depression.” *Nature Genetics* 50 (5): 668–81.

REVIEWERS' COMMENTS:

Reviewer #1 (Remarks to the Author):

Thank you for addressing the comments I previously raised. I have no new comments.

Reviewer #2 (Remarks to the Author):

The authors did an excellent job responding to my review and I have no further comments.